# Mast cell granule motility and exocytosis is driven by dynamic microtubule formation and kinesin-1 motor function

**Jeremies Ibanga[1], Eric L. Zhang[1], Gary Eitzen[1,2]\*, Yitian Guo[2]**

**1** Department of Cell Biology, University of Alberta, Edmonton, Alberta, Canada, **2** Department of Medicine, University of Alberta, Edmonton, Alberta, Canada

\* Gary.Eitzen@ualberta.ca

## Abstract

Mast cells are tissue-resident immune cells that have numerous cytoplasmic granules which contain preformed pro-inflammatory mediators. Upon antigen stimulation, sensitized mast cells undergo profound changes to their morphology and rapidly release granule mediators by regulated exocytosis, also known as degranulation. We have previously shown that Rho GTPases regulate exocytosis, which suggests that cytoskeleton remodeling is involved in granule transport. Here, we used live-cell imaging to analyze cytoskeleton remodeling and granule transport in real-time as mast cells were antigen stimulated. We found that granule transport to the cell periphery was coordinated by de novo microtubule formation and not F-actin. Kinesore, a drug that activates the microtubule motor kinesin-1 in the absence of cargo, inhibited microtubule-granule association and significantly reduced exocytosis. Likewise, shRNA knock-down of Kif5b, the kinesin-1 heavy chain, also reduced exocytosis. Imaging showed granules accumulated in the perinuclear region after kinesore treatment or Kif5b knock-down. Complete microtubule depolymerization with nocodazole or colchicine resulted in the same effect. A biochemically enriched granule fraction showed kinesin-1 levels increase in antigen-stimulated cells, but are reduced by pre-treatment with kinesore. Kinesore had no effect on the levels of Slp3, a mast cell granule cargo adaptor, in the granule-enriched fraction which suggests that cargo adaptor recruitment to granules is independent of motor association. Taken together, these results show that granules associate with microtubules and are driven by kinesin-1 to facilitate exocytosis.

## Introduction

Mast cells are tissue-resident immune cells localized in epithelial and mucosal tissues that are in contact with the external environment such as the respiratory tract, gastrointestinal tract and skin [1]. Mast cells are morphologically characterized by the presence of numerous cytoplasmic granules which house pro-inflammatory mediators, such as histamine and proteinases and molecularly characterized by the presence of FcεR1, the high affinity receptor for IgE. Upon activation, mast cells undergo profound changes in cell morphology and granule exocytosis,

**Funding:** J. I. is the recipient of an Undergraduate Student Research Award (USRA) from The Natural Sciences and Engineering Research Council of Canada (https://www.nserc-crsng.gc.ca/index_eng. asp). E.L.Z. is the recipient of a studentship from the Canadian Institutes of Health Research (https:// cihr-irsc.gc.ca/e/193.html). Y. G. is the recipient of a studentship from the China Scholarship Council (https://www.chinesescholarshipcouncil.com). This work was supported by a Discovery Grant from The Natural Sciences and Engineering Research Council of Canada (RGPIN-2019-05466) to G.E. The funders had no role in study design, data collection and analysis, decision to publish, or preparation of the manuscript.

**Competing interests:** The authors have declared that no competing interests exist.

also known as degranulation, is rapidly triggered. Degranulation is the release of granule contents by regulated exocytosis into the interstitial space [2]. Mast cell degranulation gives rise to inflammation which plays an important role in immune defense but also plays a role in the pathophysiologies of diseases such as allergic asthma and autoimmune disease [3, 4].

Antigen binding to IgE bound FcεR1 induces receptor aggregation which leads to a signaling cascade involving activation of membrane proximal receptor tyrosine kinases [5, 6], downstream Rho GTPases [7] and transient increase in intracellular calcium concentration [8, 9]. Calcium transients in conjunction with Rho GTPases modulate cytoskeletal arrangements which facilitate granule exocytosis [10, 11]. Calcium can act on multiple targets that directly activate actin and microtubule remodeling in mast cells. It has been shown that calcium activates gelsolin and cofilin that sever existing filaments, allowing turnover and the formation of new filaments [12, 13]. Calcium is required for the organization of microtubule protrusions at the periphery of mast cells [14]. Rho GTPases assemble complexes that generate new F-actin structures: activated Rac1 stimulates the formation of lamellipodia and these protrusions can be seen at the periphery of activated mast cells, while activated RhoA stimulates the formation of stress fibers that establish polarized protrusions where granules traffic [7, 15].

While actin remodeling facilitates the generation of morphological transitions during mast cell stimulation [7], it has been shown that granule transport depends on microtubule dynamics [16, 17]. Kinesin-1 is a microtubule motor protein that transports anterograde cargo to the cell periphery; it was shown in a variety of cell types to be involved in transport of secretory cargo to the cell periphery for exocytosis [18–20]. Cargo association with kinesin-1 activates its motor activity and thus movement of granules on microtubules is controlled by cargo adaptors, a large family of proteins that link vesicle transport to cellular signaling. In mast cells, it was recently shown that granules recruit the cargo adaptor, Slp3, via a mechanism that involves signaling through Rab27 [20].

Studies to show the role of cytoskeletal remodeling in cellular processes has been greatly facilitated by the availability of small-molecule inhibitors that perturb actin and microtubule dynamics [21]. Jasplakinolide and paclitaxel are drugs that directly bind to F-actin and microtubule filaments, respectively, which results in the stabilization of these cytoskeletal elements. Stabilization of the cytoskeleton has shown a slight reduction in mast cell exocytosis [22]. Latrunculin and nocodazole or colchicine are drugs that directly bind to actin and tubulin monomers, respectively, which results in a net depolymerization of these cytoskeletal elements. Mast cell granule trafficking is dependent on microtubules and exocytosis is inhibited by microtubule depolymerizing agents [23, 24]. F-actin depolymerization agents have shown variable results but typically cause a slight increase in granule exocytosis [7, 16, 22]. Several studies have shown a role for functional coordination between the remodeling of F-actin and microtubules via Rho signaling proteins [25–27].

While there is clear evidence of microtubules driving granule exocytosis, the mechanism and molecular machinery involved in recruiting the kinesin motor to mast cell secretory granules is still not understood [28, 29]. To investigate this further, we studied the effect of the small molecule, kinesore, on granule exocytosis. Kinesore is a new drug that targets kinesin-cargo adaptor function, resulting in aberrant remodeling of the microtubule network and the loss of directional vesicle transport [30].

Previously, we have shown that Rho proteins are involved in mast cell degranulation using a combination of genetic deletion [15] and acute inhibition with Rho specific drugs [7]. Here, we wanted to examine how mast cell-regulated exocytosis involves distinct cytoskeletal dynamics. We used the cytoskeletal probes Lifeact-GFP and EB3-GFP, which bind F-actin and microtubule ends respectively, to image dynamic cytoskeletal remodeling during mast cell activation. We have previously shown that mast cell activation for pro-inflammatory functions

results in cytoskeletal induced changes in cell morphology that increase cell adhesion and create zones of exocytosis which appear as cellular projections driven by F-actin remodeling. However, here we show that transport of granules into exocytosis zones requires dynamic microtubule formation and kinesin-1 motor function. Disruption of microtubule dynamics or kinesin motor transport, effectively disrupted granule transport to the cell periphery and subsequent exocytosis.

## Materials and methods

### Cells and cell culture

RBL-2H3 cells, obtained from ATCC (Manassas, VA, USA), were grown as monolayer cultures in minimal essential medium Eagles (MEM) supplemented with 10% heat-inactivated fetal bovine serum (FBS), 100 U/ml penicillin, and 100 μg/ml streptomycin at 37˚C and 5% $CO_2$. Depletion of the kinesin-1 heavy chain subunit, Kif5b, in RBL-2H3 cells was by RNA interference (RNAi) using MISSION lentiviral shRNAs (Sigma) from the mouse TRC1 library (TRC_ID: TRCN0000091[xxx], [xxx] = 479, 480 or 481). Mouse sequences were either identical to rat or contained one miss-match. Murine bone marrow-derived mast cells (BMMCs) were isolated as previously described [15]. BMMCs were cultured in RPMI media supplemented with 10% heat-inactivated FBS, non-essential amino acids (Gibco) and sodium pyruvate at 37˚C and 5% $CO_2$. Growth for four weeks in the presence of 20 ng/ml interleukin-3 was used to induce differentiation into mast cells. Maturation of mast cells was confirmed by FcεRI expression via flow cytometry. To test the effects of drugs, cells were pre-incubated for 30 min in 1 μM nocodazole (Sigma), 10 μM colchicine (Tocris) and 10 μM paclitaxel (Sigma), which target microtubules; and 100 μM kinesore (Tocris), which targets the microtubule motor kinesin-1. Drugs were dissolved in DMSO at 10 mM except kinesore which was dissolved at 20 mM; vehicle controls used DMSO at 0.5% (v/v).

**Mast cell stimulation and exocytosis assay.** Mast cells were sensitized by incubation with 120 ng/ml anti-DNP-IgE (Sigma-Aldrich) for 4 h and then stimulated by replacing the media with HEPES-Tyrode's buffer (HTB: 25 mM HEPES, pH 7.4, 120 mM NaCl, 5 mM $MgCl_2$, 1.5 mM $CaCl_2$, 1 g/L glucose, and 1 g/L BSA) containing 25 ng/ml DNP$_{\sim30}$-BSA (ThermoFisher). Unstimulated cells (i.e. resting cells) were treated with HTB without DNP-BSA. The release of β-hexosaminidase from RBL-2H3 cells and BMMCs was used to measure exocytosis/degranulation. RBL-2H3 cells were plated in 24-well plates at a density of 100,000 cells/well, and BMMCs at 300,000 cells/well and grown overnight. After IgE sensitization, cells were pretreated with the indicated drugs or DMSO (vehicle) for 30 min at 37˚C. Cells were then stimulated with 25 ng/ml of DNP-BSA for 30 min. After stimulation, samples were placed on ice to stop exocytosis, supernatant were centrifuged at 5000*g* for 5 min to remove residual cells, then 100 μL of cell-free supernatant was incubated with 100 μL of 1.2 mM 4-methylumbelliferyl N-acetyl-β-D-glucosaminide (MUG) (Sigma) in 50 mM sodium citrate pH 4.5 for 30 min at 37˚C. The reaction was terminated by the addition of 50 μL of 0.1 M glycine (pH 10). Cleavage of MUG by β-hexosaminidase releases the fluorescent product 4-methylumbelliferone, which was detected with a Synergy-4 fluorometer set to 360 nm +/- 20 nm excitation and 450 nm +/-20 nm emission (BioTek Instruments). Fluorescence is directly proportional to exocytosis, which was calculated as the percentage of β-hexosaminidase activity in the supernatant, divided by total β-hexosaminidase activity as determined from Triton X-100 lysed cells.

### Protein preparation and granule enrichment

Granule-enriched fractions were prepared by differential centrifugation of RBL-2H3 cells lysed in 0.3 M sucrose to osmotically stabilize granules [31]. RBL-2H3 cells were grown to 80%

confluence in 10 cm dishes, then pre-treated with 100 μM kinesore or 0.5% (v/v) DMSO for 30 min followed by stimulation or left unstimulated. Cells were washed with 10 mM HEPES pH 7.5, 0.3 M sucrose, 0.5 mM PMSF, 1 μg/ml leupeptin, 0.5 μg/ml pepstatin and frozen in 1 ml of the same buffer. Cells were thawed and scraped from plates, then passed 12 times through a ball bearing homogenizer with a 12-micron gap. Whole cell lysates were obtained after centrifugation at 450*g* for 15 min at 4˚C to remove unbroken cells. Granule and mitochondria enriched fractions were obtained after centrifugation of lysates at 4000*g* for 20 min at 4˚C. Membrane fractions were washed in 10 mM HEPES pH 7.5, 0.3 M sucrose and centrifuged again. Membrane pellets showed no contamination with cytosol as determined by immunoblot for β-tubulin which was detected only in supernatant fractions. Lysates and fractions were probed by immunoblot for the presence of Kif5b using rabbit anti-Kif5b antibody (Proteintech Group), the cargo adaptors Slp3 using rabbit anti-SYTL3 antibodies (Proteintech Group) and SKIP using rabbit anti-PLEKHM2 antibodies (Novus), the mitochondrial marker ATPIF1 using mouse monoclonal anti-ATPIF1 antibody (clone 5E2D7, Thermo Fisher), and the granule marker RMCP II using rabbit anti-RMCP II antibodies (a gift from D. Befus, University of Alberta; obtained from H.R.P. Miller, University of Edinburgh [32]). Secondary antibodies were Alexa Fluor 680 goat anti-rabbit and Alexa fluor 770 goat anti-mouse (Thermo Fisher). Blots were scanned and quantified using an Odyssey far red imager (LiCor Inc.).

## Microscopy

Fixed-cell fluorescence microscopy was performed on RBL-2H3 cells to visualize F-actin, microtubules, and granule distribution. Cells were pre-treated with the respective drug or vehicle for 30 min at 37˚C, except kinesore pre-treatment in Fig 5B was for the indicated times. Cells were then left unstimulated or stimulated with 25 ng/ml of DNP-BSA for 30 min at 37˚C. Cells were fixed for 20 min with 4% (wt/v) paraformaldehyde prior to and after stimulation. F-actin was labelled with Oregon green 488 phalloidin (Thermo Fisher) or tubulin was labelled with rabbit anti-β-tubulin antibodies (Abcam), granules were labelled with mouse monoclonal anti-CD63 antibodies (clone AD1, BioRad) and nuclei labelled with DAPI. Labelled secondary antibodies used were Alexa Fluor 555 donkey anti-mouse and Alexa Fluor 488 donkey anti-rabbit (Thermo Fisher). Cells were imaged using 63X objective (1.4 NA) on a Zeiss Observer Z1 epifluorescence microscope and processed with Axiovision v4.8 software.

Live-cell fluorescence microscopy was performed on RBL-2H3 cells during stimulation to visualize dynamic change in cell morphology, via cytoskeletal remodeling and granule movement. RBL-2H3 cells were transfected by electroporation (250 V, 950 μF), as previously described [9], with 8 μg of Lifeact-GFP [33], EB3-GFP [34] or EB3-tdTomato [35] plasmids. After transfection cells were plated onto 25 mm round coverslips and grown overnight, then sensitized with anti-DNP-IgE for 4 h and granules were stained using 500 nM LysoTracker Red or LysoTracker Green (Thermo Fisher). Coverslips were placed in an Attofluor chamber and growth media was replaced with HTB. After 1 min of imaging, resting cells were stimulated by the addition of 25 ng/ml DNP-BSA and drugs or DMSO were added at the same time. Imaging was done on a Leica UltraVIEW VoX spinning-disk confocal microscope (PerkinElmer) using a 63X (1.4 NA) objective. Images were taken every 10 s and processed with Volocity 6.1 imaging software. Images were compiled and exported as videos on windows media at 10 frames/s.

## Image analysis

Granule motility was determined by particle tracking software in Volocity 6.1. The velocities of LysoTracker-labelled particles between 50 and 200 μm diameter were determined if they

could be continuously tracked through 3 or more frames. Hence, velocities were calculated only from moving particles near the cell periphery as some LysoTracker-labelled particles do not move and perinuclear particles were excluded due to size constraints.

To determine the number of EB3 puncta in cells expressing EB3-GFP, frames from live cell imaging time-courses were extracted and the threshold was set on the GFP channel to exclude 99% of the signal leaving only microtubule ends and the microtubule organizing center visible. ImageJ was used to quantify the number of particles between 4 to 50 square pixels. An example of this analysis is shown in *Supporting information* (S1 Fig).

The distribution of CD63 signal in cells, as examined by profile plots, was used to indicate whether a specific condition affected granule motility. RBL-2H3 cells were triple labelled for CD63 (red), cytoskeleton (green) and nuclei (blue). Profile plots of the distribution of signal intensity across a cross-section of a cell that included the nucleus and the cell periphery were generated using ImageJ. To quantify granule distribution after antigen-stimulation, a region of interest was drawn around the perinuclear region using a 2 μm border (ROIpn) or around the perimeter of the whole cell (ROIwc). The amount of CD63 signal in the ROIpn and the ROIwc was quantified in ImageJ as integrated density for 10 cells per condition, and values were calculated for the percentage of signal in the ROIpn ((ROIpn/ROIwc)*100%).

### Gene expression by quantitative PCR

Total RNA was isolated from 3 x 10⁶ RBL-2H3 cells using TRIzol reagent (Thermo Fisher). 5 μg of total RNA was reversed-transcribed into cDNA using SuperScript II Reverse Transcriptase (Invitrogen) using 50 μM oligo dT, per the manufacturer's instructions. Levels of *KIF5B*, *JIP1*, *JIP2*, *JIP3*, *JIP4*, *SYTL1*, *SYTL2*, *SYTL3(SLP3)*, *SYTL4*, *SYTL5*, *PLEKHM1*, *SKIP (PLEKHM2)*, were determined via quantitative PCR assays using SensiFAST SYBR No-ROX reagent (Bioline). Reactions were carried out in triplicate using 100 ng of cDNA per reaction. Thermocycling used an Eppendorf Mastercycler ep realplex in a two-step PCR (95°C, 5 s then 62°C, 20 s for 40 cycles). Relative mRNA levels were quantified using the delta-delta CT method(2^-ΔΔCT) and normalized against the mean values of GAPDH acting as an endogenous control. Table 1 indicates the primer sequences and amplicon lengths.

### Statistical analysis

Values reported are the statistical mean +/- standard error of the mean. We used an unpaired, two-tailed Student's *t*-test to determine if the differences between two datasets was statistically significant (*p-value*).

## Results

### Live-cell imaging of cytoskeletal dynamics during mast cell stimulation implicates a role for microtubules in granule exocytosis

We recently showed that drugs targeting Rho proteins inhibit mast cell granule exocytosis and morphological transition into an activated state induced by stimulation via FcεRI aggregation [7]. Rho proteins control actin remodeling which seems to play a role in secretory granule transport and exocytosis in many cell types [36–38] including mast cells (*reviewed in* Ménasché et al., 2021 [39]). To examine this in secretory granule-rich mast cells, we used the F-actin probe, Lifeact-GFP [33], to microscopically image the dynamics of actin remodeling during antigen-stimulation. RBL-2H3 cells, an adherent cell line typically used as a model of mast cells [40], were transfected with the Lifeact-GFP probe and granules were labelled with LysoTracker red to examine the coordination between new F-actin

**Table 1. qPCR primers for Kinesin-1 cargo adaptor proteins.**

| Gene | Accession | Primers 5' Forward 3' Reverse | Sequence (5'-3') | Amplicon Length |
|---|---|---|---|---|
| Styl1 | NM_001025651.1 | 5' Styl1 Rat qPCR p417 | CGAAGGAAGAAGAGCTCTAAGG | 88 |
| | | 3' Styl1 Rat qPCR p505 | CTGCCTCTTCTATGGTGTCTTC | |
| Styl2 | NM_001108492.2 | 5' Styl2 Rat qPCR p2027 | CCGATGTCTTCTGGGTCTATTC | 101 |
| | | 3' Styl2 Rat qPCR p2128 | TCACTGATGATGGATGAACTCTC | |
| Styl3 | NM_001127560.1 | 5' Styl3 Rat qPCR p537 | AACCGGAGAGTGGTTCTTTG | 104 |
| | | 3' Styl3 Rat qPCR p641 | TGCTCTGCCTCTGATAAGATTG | |
| Styl4 | NM_080410.1 | 5' Styl4 Rat qPCR p1585 | CTCCAGGTGTGGATCAAAGAAG | 97 |
| | | 3' Styl4 Rat qPCR p1682 | TCCTCATGGGAAGGAGGTATC | |
| Styl5 | NM_178333.1 | 5' Styl5 Rat qPCR p99 | GGTGGAAGACAAGAGGATAAGG | 108 |
| | | 3' Styl5 Rat qPCR p207 | TTCTGACAGTGAACACAGACTC | |
| Plekhm1 | NM_001009677.1 | 5' plekhm1 Rat qPCR p2722 | TCATAACTGGGACCTCACAAAG | 128 |
| | | 3' plekhm1 Rat qPCR p2850 | TGCGCTCTACATGCTCATAC | |
| Plekhm2 (SKIP) | NM_001191767.1 | 5' SKIP Rat qPCR p2648 | CCGTATCCAAAGGAGTCATACC | 100 |
| | | 3' SKIP Rat qPCR p2748 | TCTTCGTGGCATGTGAAGAG | |
| JIP1 | NM_053777.1 | 5' JIP1 Rat qPCR p2860 | TGTGTGTTCAAAGAAGGAGAGG | 108 |
| | | 3' JIP1 Rat qPCR p2968 | AGTAGTAGCCAGGTGACAAGA | |
| JIP2 | NM_001100720.1 | 5' JIP2 Rat qPCR p2547 | CTGTGTGGTCAATGGAGAAGAG | 99 |
| | | 3' JIP2 Rat qPCR p2646 | GGGTCGTCCACATCTAACTCTA | |
| JIP3 | NM_001100673.1 | 5' JIP3 Rat qPCR p1679 | AGTTCTTTAGCCGCCTCTTC | 96 |
| | | 3' JIP3 Rat qPCR p1775 | CTGTAGTGGGTGACTTGTAGTG | |
| JIP4 (Spag9) | NM_001108290.2 | 5' JIP4 Rat qPCR p2398 | CAGCACCCATTCAACTACAAAG | 103 |
| | | 3' JIP4 Rat qPCR p2501 | GGCAATGCACAGAACATGAG | |

formation and granule transport. Live-cell imaging showed the formation of actin rich lamellipodia approximately 5 min after antigen stimulation (Fig 1A, *5 min*; see **S1 Video** via https://doi.org/10.6084/m9.figshare.19349552.v1). However, granules did not appear in these actin-rich protrusion as they formed (Fig 1B, *asterisk*); instead granules showed a delay in moving into the protrusions (Fig 1B, *arrow*). There was no observable overlapping signal from F-actin and granules. This suggests that F-actin may not facilitate granule transport to the cell periphery.

Next, we examined microtubule dynamics during mast cell antigen-stimulation. RBL-2H3 cells were transfected with EB3-GFP, which labels the growing ends of microtubules [34], and granules were labelled with LysoTracker red. New microtubules radiated from a juxtanuclear position within the cell, likely representing the microtubule organizing center (Fig 1C, **S2 Video** via https://doi.org/10.6084/m9.figshare.19349555.v1). During stimulation granules were observed to project in conjunction with microtubule growth to the cell periphery. Granules at the cell periphery moved in a coordinated manner similar to the growth of EB3-GFP positive microtubules (Fig 1D). The velocity at which EB3-GFP puncta and Lyso-Tracker red-labelled granules moved was determined using particle tracking software. This showed that microtubules grew at 0.57 +/-0.033 µm/s, which is similar to previously determined rates in COS-1 cells [34]. Granules that were tracked at the cell periphery moved at 0.46 +/-0.22 µm/s. Hence granules moved at velocities similar to the rate of microtubule growth which may facilitate coordination between granules motility and de novo microtubule formation. Many LysoTracker labelled granules in the interior of the cell did not move and were excluded from granule velocity calculations since they likely represent a pool not activated for exocytosis.

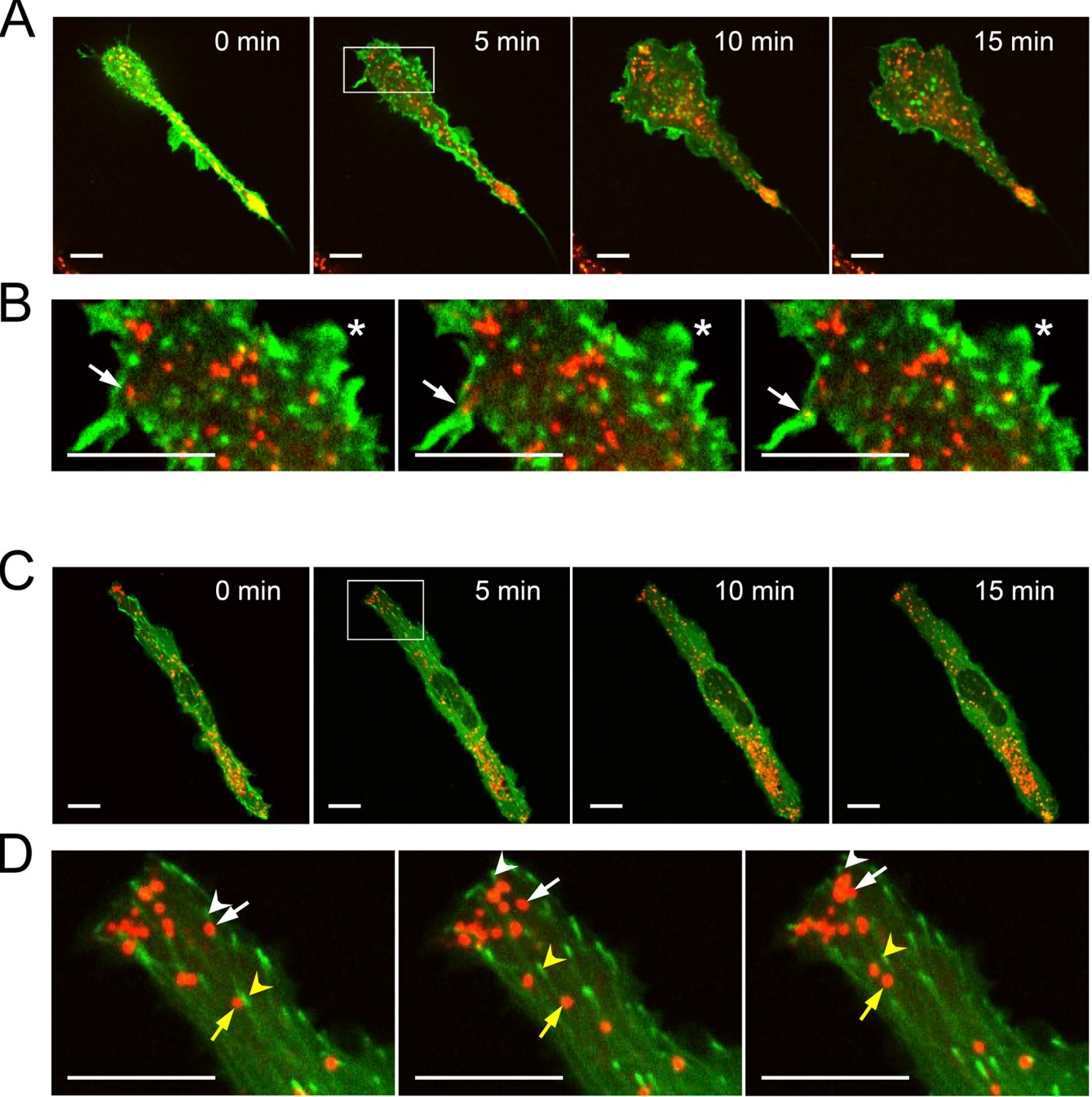

**Fig 1. Live-cell imaging of cytoskeletal dynamics and granule movement during stimulation of mast cells.** RBL-2H3 cells were transfected with Lifeact-GFP (*A and B*) or EB3-GFP (*C and D*) to label nascent F-actin and microtubules respectively, and granules were stained with LysoTracker Red. **A)** Representative confocal images from se **S1 Video** (https://doi.org/10.6084/m9.figshare.19349555.v1) of F-actin remodeling and granule movement during a time course after antigen-stimulation. **B)** Magnified images from the 5 min time point at 10 s intervals showing the movement of a granule into a pre-existing actin protrusion (*arrow*), and newly forming lamellipodia devoid of granules (*asterisk*). **C)** Representative confocal images from **S2 Video** (https://doi.org/10.6084/m9.figshare.19349552.v1) of nascent microtubule formation and granule movement during a time course after antigen-stimulation. **D)** Magnified images from the 5 min time point at 10 s intervals showing the movement of granules (*arrows*) is coordinated with the growth of microtubules (*arrowheads*). Scale bar, 10 μm.

## Microtubule-targeted drugs inhibit mast cell granule exocytosis

To confirm the role of microtubules in granule transport we examined the effects of microtubule drugs that interfere with microtubule dynamics. Nocodazole and colchicine stimulate microtubule depolymerization while pacitaxel binds and stabilizes microtubules but also arrests their growth [41]. Nocodazole potently inhibited granule exocytosis (Fig 2A; [ref. 7]). We also tested colchicine as an alternative microtubule depolymerizing agent, which also inhibited exocytosis, although it was less potent than nocodazole (Fig 2A). Exocytosis was only partially affected by paclitaxel treatment, with a 30% drop in levels at the highest concentration (Fig 2A).

We used live-cell imaging to further characterize the dynamic effects of microtubule drugs. RBL-2H3 cells were transfected with EB3-GFP to label nascent microtubules and granules were co-stained with LysoTracker red. Live cell imaging of nocodazole and colchicine treated RBL-2H3 cells showed EB3-GFP puncta rapidly dissolved. However, granules still maintained some motility and antigen stimulation resulted in transient peripheral protrusions that were likely F-actin based since they resembled lamellipodia (see **S3 Video** via https://doi.org/10. 6084/m9.figshare.19349558.v1 and **S4 Video** via https://doi.org/10.6084/m9.figshare. 19349564.v1). Still images at different time points showed changes in granule distribution (Fig 2B and 2C). Paclitaxel treatment rapidly froze microtubule dynamics and caused a slow loss of EB3-GFP puncta; no projections formed and granule movement was static (see **S5 Video** via https://doi.org/10.6084/m9.figshare.19349567.v1). Images from different time points after pac- litaxel treatment showed little change in granule distribution (Fig 2D). Particle tracking analy- sis of granules showed that paclitaxel reduced granule motility by 52%, while nocodazole and colchicine did not have a significant effect (Fig 2E). This is a seemingly inconsistent effect since nocodazole and colchicine reduced exocytosis to a much greater extent than paclitaxel. However, granule movement after microtubule depolymerization was no longer directional to the cell periphery but remained as stochastic movements. Our particle tracking analyzed did not take trajectories into consideration. The reduction of granule movement on paclitaxel-sta- bilized microtubules likely due to granules that remained fixed to this static network.

## The microtubule motor modulator drug, kinesore, inhibits granule trafficking and exocytosis

Based on the effects of microtubules drugs, granule motility to the cell periphery and subse- quent exocytosis seems to be dependent on microtubule dynamics. This observation concurs with work previously demonstrating microtubules are required for granule motility in bone marrow-derived mast cells (BMMCs) and RBL-2H3 cells [16, 17, 23, 24]. However, microtu- bules are required for many cellular functions, hence drugs that target microtubules might affect exocytosis indirectly. To directly investigate granule trafficking via microtubule motor proteins, we examined the effect of a recently identified drug, kinesore, a small molecule com- pound that binds and activates the microtubule motor protein, kinesin-1, in the absence of cargo binding [30]. Kinesore pre-treatment showed a dose-dependent inhibition of mast cell exocytosis in both RBL-2H3 cells and BMMCs (Fig 3A). The reduction in signal was 41 +/-2.3% for BMMCs and 68 +/-10.4% for RBL-2H3 cells. This may seem modest, however, we used kinesore at concentrations up to its $IC_{50}$ of 100 μM for blocking interactions with the cargo adaptor SKIP [30]; therefore, an approximate 50% reduction in signal would be expected.

We performed live-cell imaging of antigen-activated RBL-2H3 cells to characterize the dynamic effects of kinesore on cytoskeletal remodeling and granule motility. EB3-tdTomato and LysoTracker green were used to visualize microtubule dynamics and granule movement,

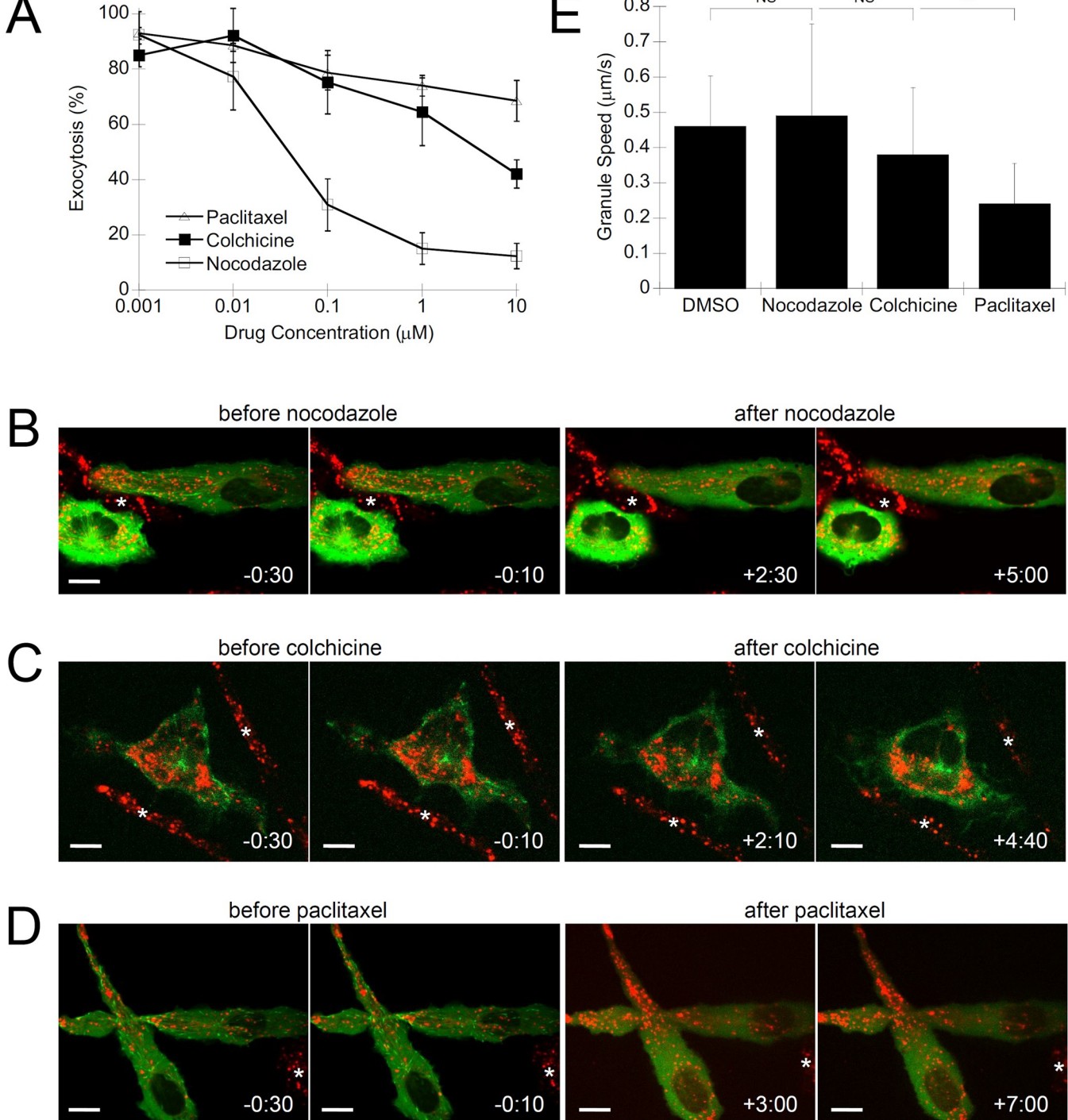

**Fig 2. Microtubule drugs inhibit mast cell granule exocytosis and affect granule motility. A)** Exocytosis assay of RBL-2H3 cells that were pre-treated with nocodazole, colchicine and paclitaxel for 30 min prior to antigen-stimulation. Exocytosis was assayed as the percent β-hexosaminidase released of total, normalized to vehicle (DMSO) controls (n = 3). **B–D)** Representative confocal images from live-cell imaging of microtubule dynamics and granule movement of mast cells treated with 1 μM nocodazole (*B*, **S3 Video** via https://doi.org/10.6084/m9.figshare.19349558.v1), 10 μM colchicine (*C*, **S4 Video** via https://doi. org/10.6084/m9.figshare.19349564.v1), or 10 μM paclitaxel (*D*, **S5 Video** via https://doi.org/10.6084/m9.figshare.19349567.v1). RBL-2H3 cells were transfected with EB3-GFP to label nascent microtubules and incubated with LysoTracker red to label granules. Cells were imaged for 1 min, then antigen-stimulated and concurrently drugs were added, followed by 15 min of imaging. Scale bar, 10 μm; * indicates untransfected cells. **E)** Granules at the cell periphery were analyzed by particle tracking software. A minimum of 15 granules from 5 cells were tracked. Shown is the mean granule speed +/- standard error. The effect of drugs was compared to the vehicle (0.5% DMSO) control by Student's *t*-test (NS, not significant; **, *p* = 0.0074; n = 5).

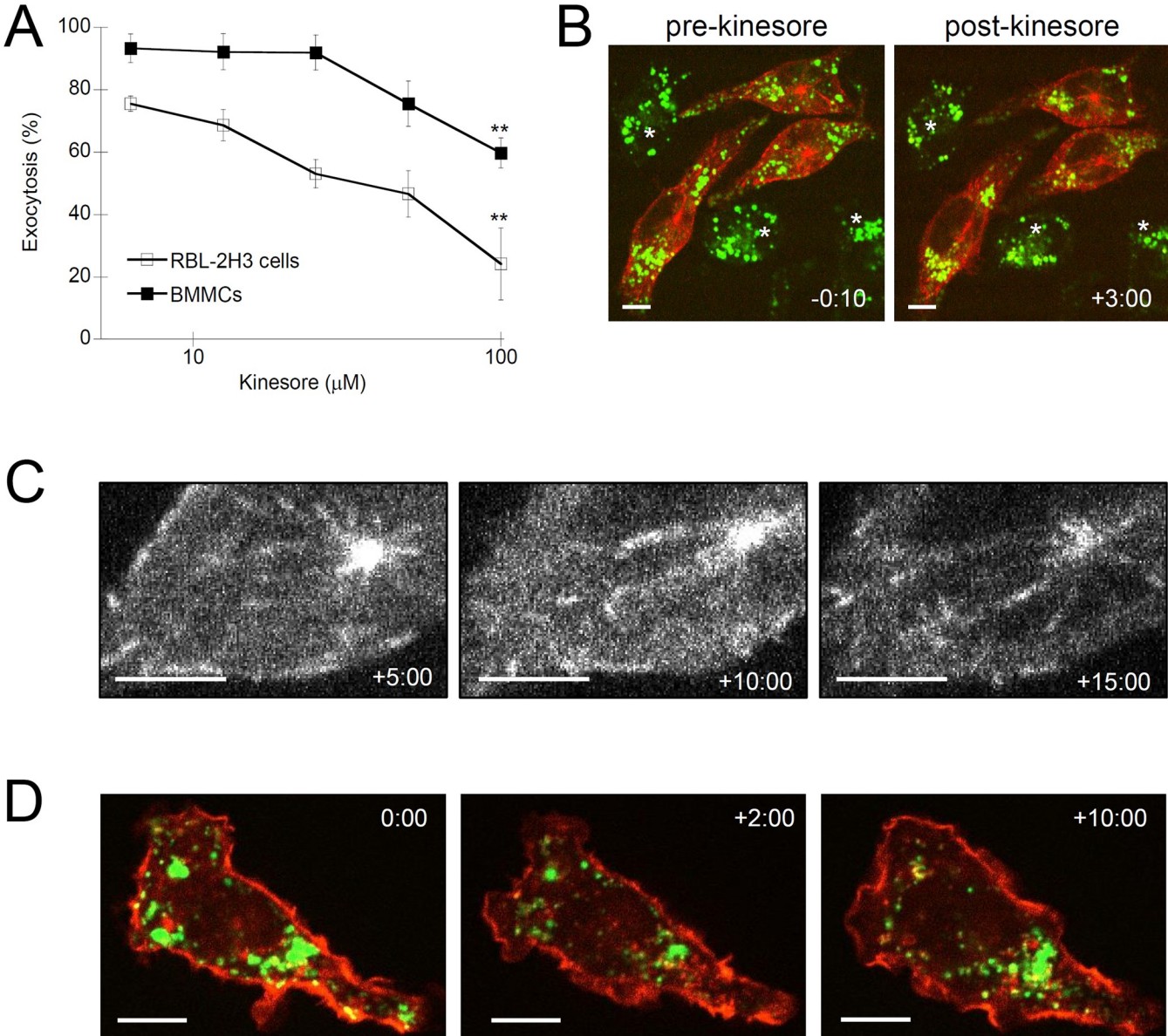

**Fig 3. Kinesore, a small-molecule activator of microtubule motors, inhibits mast cell granule exocytosis. A)** Exocytosis assay of RBL-2H3 cells and BMMCs that were pre-treated with kinesore for 30 min prior to antigen-stimulation. Exocytosis was assayed as the percent β-hexosaminidase released of total, normalized to vehicle (DMSO) controls. 100 μM kinesore showed statistically significant inhibition of both RBL-2H3 cell and BMMC exocytosis via Student's *t*-test (**, *p* = 0.0104 and 0.0088 for RBL-2H3 and BMMC respectively; n = 3). **B)** Representative confocal images from live-cell imaging of microtubule dynamics and granule movement of mast cells treated with 100 μM kinesore (see **S6 Video** via https://doi.org/10.6084/m9.figshare.19349573.v1). RBL-2H3 cells were transfected with EB3-tdTomato to label nascent microtubules and incubated with LysoTracker green to label granules. Cells were imaged for 1 min, then antigen-stimulated and concurrently 100 μM kinesore was added, followed by 15 min of imaging showing the movement of granules to the perinuclear region while microtubules project to the cell periphery. **C)** Magnified images from the 5 min– 15 min time points showing the formation of EB3-tdTomato puncta are not affected by kinesore. **D)** Representative confocal images from live-cell imaging of actin dynamics and granule movement of mast cells treated with 100 μM kinesore (see **S7 Video** via https://doi.org/10.6084/m9.figshare.19349579.v1). RBL-2H3 cells were transfected with Lifeact-Ruby to label nascent F-actin and incubated with LysoTracker green to label granules. Cells were imaged for 1 min, then antigen-stimulated and concurrently 100 μM kinesore was added followed by 15 min of imaging. Scale bar, 10 μm; * indicates untransfected cells.

respectively (see **S6 Video** via https://doi.org/10.6084/m9.figshare.19349573.v1). Still images showed that granules were at first spread throughout the cell; however, after kinesore exposure granules no longer projected to the cell periphery and instead accumulated in the perinuclear region (Fig 3B). EB3-tdTomato labelling of the microtubule network was not affected by kinesore (Fig 3C). A similar number of EB3-tdTomato puncta occurred in cells over the time course of kinesore treatment (S1 Fig). Lifeact-Ruby and LysoTracker green were used to visualize F-actin remodeling and granule movement after kinesore treatment (see **S7 Video** via https://doi.org/10.6084/m9.figshare.19349579.v1). Still images of Lifeact-Ruby-labelled F-actin showed that kinesore had no effect on actin remodeling (Fig 3D). Membrane ruffling, an actin remodeling induced phenomena observed activated mast cells, was consistently observed in antigen-activated RBL-2H3 cells pretreated with kinesore. These results support a role for microtubule-directed granule motility and suggest that kinesin motors play a role in the transport of secretory granules.

### Microtubule-targeted drugs do not block cellular transition to activated phenotype

We next used immunofluorescence microscopy to examine the effect of microtubule drugs and the microtubule motor modulator drug, kinesore, on cytoskeletal structures and the distribution of mast cell granules labelled with anti-CD63 antibodies (CD63+). RBL-2H3 cells were pre-treated with vehicle (0.5% DMSO), nocodazole, colchicine, paclitaxel or kinesore, then left unstimulated or antigen-stimulated for 30 min, fixed and labelled for immunofluorescence microscopy. Vehicle-treated cells showed typical morphology; unstimulated cells were elongated with CD63+ granules enriched in the perinuclear region (Fig 4A) while stimulated cells showed cell spreading and peripheral lamellipodia formation with CD63+ granules spread throughout the cell (Fig 4A'). Unstimulated nocodazole and colchicine-treated cells showed retraction with granule staining that was intensely perinuclear (Fig 4B and 4C). However, stimulated cells pre-treated with colchicine and nocodazole underwent normal cell spreading and formed peripheral lamellipodia; however, granules remained perinuclear which was much more apparent in nocodazole-treated cells (Fig 4B' and 4C'). Unstimulated paclitaxel-treated cells were elongated and showed actin ruffles (Fig 4D); after stimulation, actin ruffles persisted and peripheral lamellipodia formed but there was a lack of C63+ granules at the cell periphery (Fig 4D'). Interestingly, kinesore-treated cells showed the formation of some peripheral lamellipodia prior to stimulation (Fig 4E, *arrow*). After stimulation, kinesore-treated cells transitioned to an activated morphology; cells spread and flattened however CD63+ granules did not redistribute throughout the cell and instead accumulated in the perinuclear region (Fig 4E'). Profile plots show the spread of CD63+ signal throughout vehicle-treated stimulated cells, while in drug-treated cells the CD63+ signal was predominantly adjacent to the nucleus signal (Fig 4, *left panels*).

   The distribution of fluorescence signal in the indicated cross-sections a cells was analyzed by profile plot (Fig 4, *right panels*). In unstimulated cells, profile plots showed large peaks of CD63 signal next to nuclei peak signals. (Fig 4A–4E, *right panels*). In stimulated cells that were not drug treated, the CD63 signal was distributed across the profile (Fig 4A', *right panel*). However, profile plots of stimulated cells that were drug-treated showed large peaks of CD63 signal next to nuclei (Fig 4B–4E', *right panels*). This block in the redistribution of signal throughout the cell indicates granule motility was affected. CD63 distribution was quantitatively analyzed based on the level of signal remaining in the perinuclear region after antigen-stimulation. While in control cells less than 42% of the CD63 signal remained in the perinuclear region after antigen-stimulation, drug treated cells consistently showed greater than 54% of the CD63

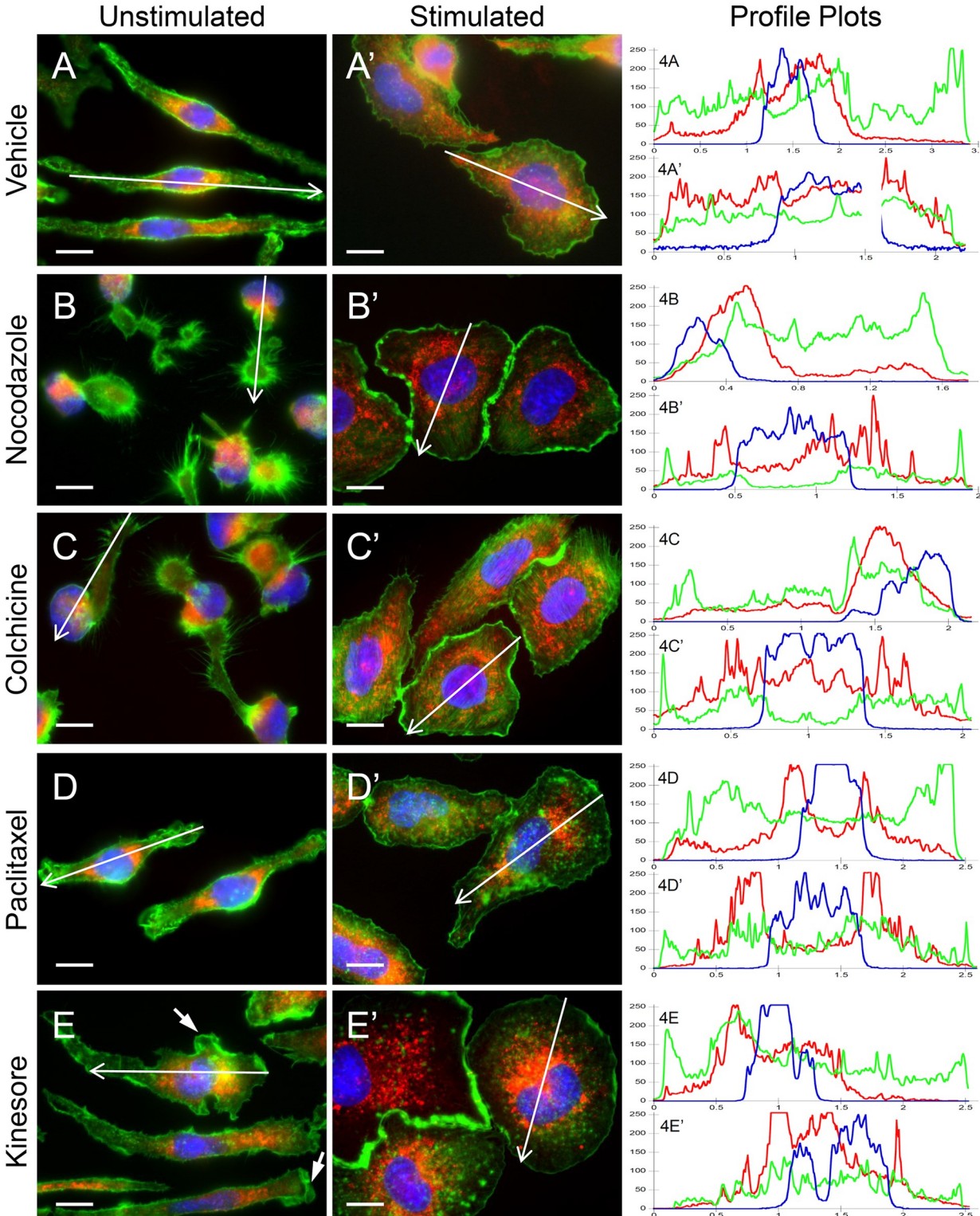

**Fig 4. Microtubule drugs and the microtubule motor drugs, kinesore, affects cell morphology and granule distribution but not F-actin remodeling.** RBL-2H3 cells were pretreated for 30 min with vehicle (*A*, DMSO), 1 μM nocodazole (*B*), 10 μM colchicine (*C*), 10 μM paclitaxel (*D*), or 100 μM kinesore (*E*). Cells were left unstimulated or antigen-stimulated for 30 min, fixed and stained for nuclei (blue), F-actin (green), or CD63 + granules (red). Scale bar, 10 μm. Profile plots (*right panels*) show the levels of CD63+ granules distributed in the indicated cross-sections. Drug pretreatment did not affect the formation of F-actin rich lamellipodia after stimulation, however CD63+ granules accumulated in the perinuclear region of drug treated cells (*B'-E'*, *red*), while control cells (*A'*, *red*) showed CD63+ granules were distributed across the entire cross-section.

**Table 2. Quantification of perinuclear-retained CD63 signal in antigen-stimulated RBL-2H3 cells.**

| Sample | CD63 signal in perinuclear region (% of total) | s.e.m. (+/-) | *t*-test vs control / scrambled |
|---|---|---|---|
| control (0.5% DMSO) | 41.8 | 4.56 | na |
| nocodazole (1 μM) | 57.2 | 4.56 | 0.0219 |
| colchicine (10 μM) | 56.2 | 6.21 | 0.0662 |
| paclitaxel (10 μM) | 68.3 | 5.00 | 0.0006 |
| kinesore (100 μM) | 66.1 | 4.50 | 0.0009 |
| scrambled (shRNA) | 47.9 | 3.60 | na |
| Kif5b KD (shRNA) | 59.7 | 3.69 | 0.0272 |

signal remained perinuclear (Table 2). Kinesore and paclitaxel showed the most striking effect with more than 66% of the CD63 signal retained in the perinuclear region. This suggests microtubules and microtubule motors drive granule translocation from the perinuclear region.

## LysoTracker colocalizes with anti-CD63 labelling

We, and others, have used LysoTracker to mark mast cell granules which facilitates their examination in living cells [7, 42, 43]. However, anti-CD63 has also been previously used to label secretory granules in mast cells and this facilitates their quantitative analysis [44, 45]. To confirm we are examining similar organelles with these labels, we examined the colocalization of LysoTracker and anti-CD63 labeling. Images showed nearly all LysoTracker-labelled vesicles colocalize with anti-CD63, although some CD63+ vesicle do not label with LysoTracker (S2 Fig). This is likely due to secretion which affects LysoTracker labelling since it requires an acidic environment. Calculation of Pearson's correlation coefficients showed a strong correlation between LysoTracker and CD63 in unstimulated cells (r = 0.714 +/-0.033) which decreased slightly in stimulated cells (r = 0.622 +/-0.055), this is likely due to the activation of exocytosis which neutralizes granules and would affect LysoTracker labelling and not CD63 (S3 Fig).

## Knock-down of Kif5b expression reduces granule exocytosis

To validate that the effects of kinesore were due to targeting kinesin-1 function, we knocked down the expression of Kif5b, the kinesin-1 heavy chain, in RBL-2H3 cells. We used three specific Kif5b shRNA and compare the effects to a mock knock-down using scrambled shRNA (Fig 5A). The Kif5b knock-down strain showed a significant reduction in antigen-stimulated exocytosis compared to both the wild-type strain and the scrambled strain (Fig 5B). Granule distribution was also affected by Kif5b knock-down. In unstimulated cells, granules resided perinuclearly (Fig 5C and 5D). In control cells, antigen-stimulation resulted in the redistribution of granules from a perinuclear region to throughout the cell (Fig 5C'). In the Kif5b knock-down strain, granules remained in the perinuclear region (Fig 5D'). Profile plots show the levels of CD63 signal in a cross-section of a cell. After stimulation, control cells showed peaks across the entire plot, while in Kif5b knock-down cells the CD63 signal was predominantly adjacent to the nucleus signal (Fig 5C' and 5D', *left panels*). CD63 distribution was quantitatively analyzed based on the level of signal remaining in the perinuclear region after antigen-stimulation. In control cells (scrambled shRNA) 48% of the CD63 signal remained in the perinuclear region, while in the Kif5b knock-down cells 60% of the CD63 signal remained

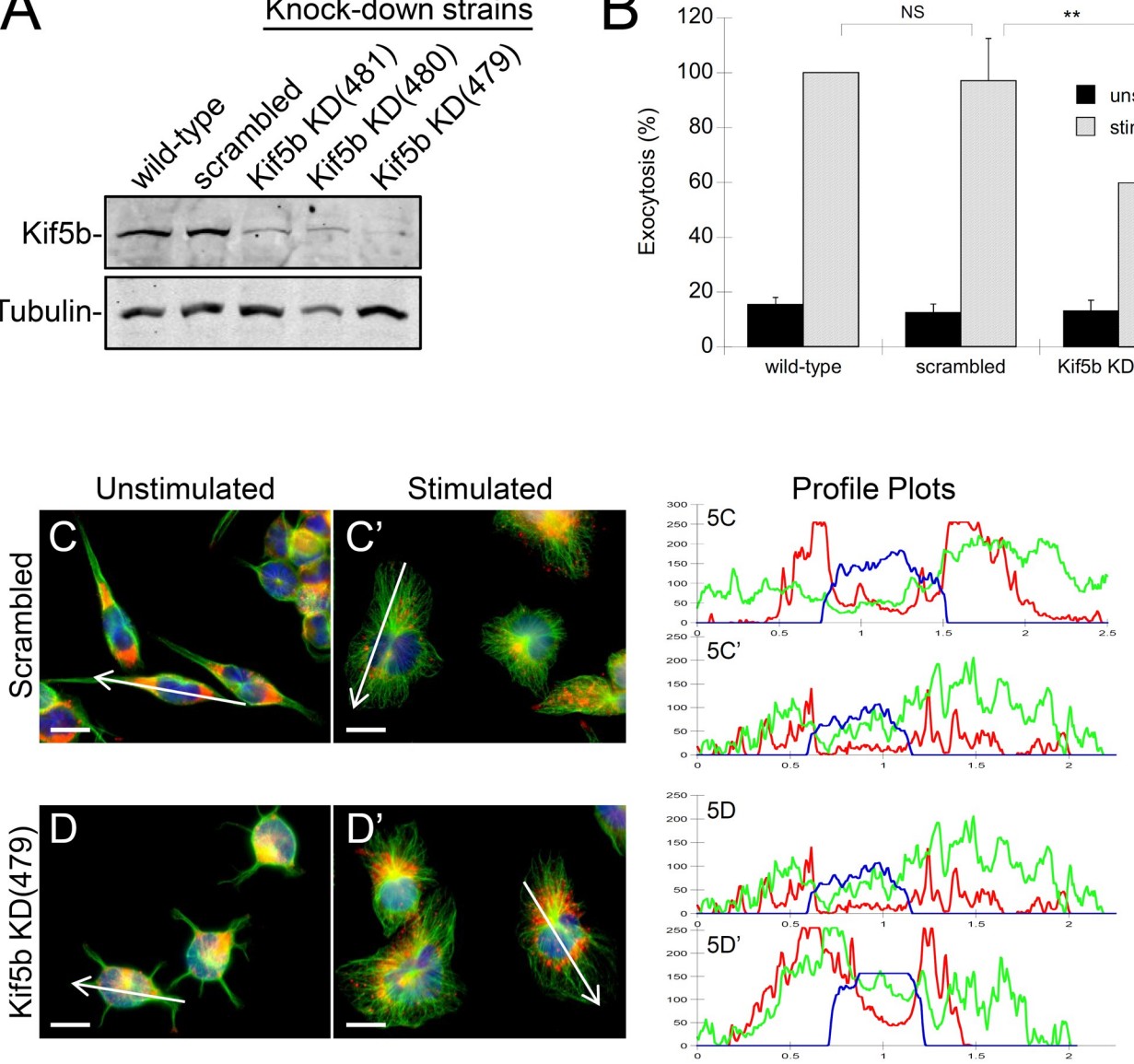

**Fig 5. Knock-down of the kinesin-1 heavy chain, Kif5b, inhibits granule translocation to the plasma membrane and exocytosis. A)** Comparison of Kif5b levels in different RBL-2H3 knock-down strains. Kif5b knock-down strains were generated with three different specific shRNAs and a non-specific scrambled shRNA as a control. Immunoblot analysis shows reduction in Kif5b in all three strains treated with specific shRNAs, with strain 479 showing the greatest reduction. Tubulin levels remained similar. **B)** Exocytosis assay of RBL-2H3 strains that were unstimulated (*black bars*), or antigen-stimulated for 30 min (*hatched bars*). The Kif5b knock-down strain showed a significant reduction in exocytosis compared to the scrambled control and wild-type strains. Exocytosis was assayed as the percent β-hexosaminidase released of total, normalized to wild-type stimulated samples. Statistical analysis was by two-tailed unpaired Student's *t*-test (NS, not significant; \*\*\*, $p = 0.0043$; \*\*, $p = 0.029$; n = 4). **C and D)** Immunofluorescence microscopy of RBL-2H3 strains showing the intracellular distribution of CD63+ granules. Control cells (*C*) or Kif5b knock-down cells (*D*) were left unstimulated or antigen-stimulated for 30 min, fixed and stained for nuclei (blue), microtubules (green), or CD63+ granules (red). Scale bar, 10 µm. Profile plots (*right panels*) show the levels of CD63 + granules distributed in the indicated cross-sections. Kif5b knock-down did not affect microtubule projections after stimulation, however CD63+ granules accumulated in the perinuclear region (*D'*, *red*), while in control cells CD63+ granules were distributed across the entire cross-section (*C'*, *red*).

perinuclear (Table 2). Note that the projection of microtubules to the cell periphery was not affected by Kif5b knock-down.

### Kinesore treatment results in microtubule looping

Microtubule morphology was also examined in kinesore-treated RBL-2H3 cells by immuno-fluorescence microscopy. Microtubules project radially from the nuclear region to the cell periphery in unstimulated and stimulated cells, with many CD63+ granules at the cell periphery after stimulation (Fig 6A, *Vehicle*). Pretreatment with kinesore resulted in microtubules that rearranged into looping structures in both unstimulated and stimulated cells (Fig 6A, *Kinesore*). This morphology is similar to that previously observed in HeLa cells treated with

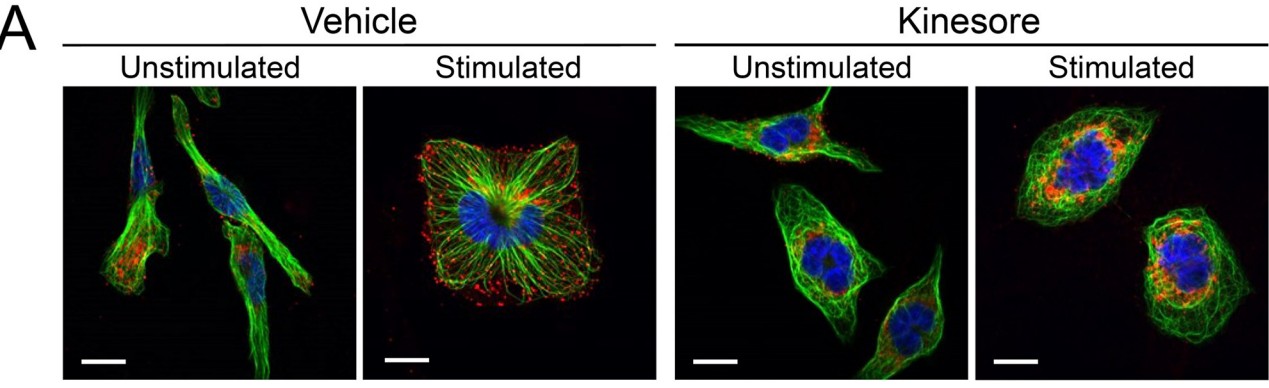

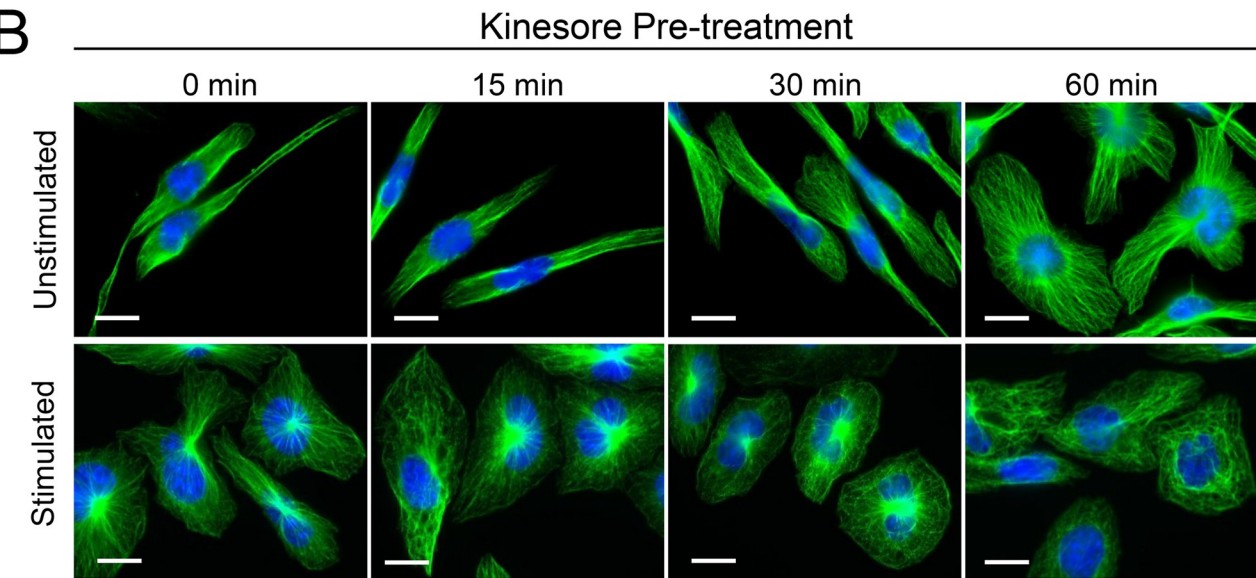

**Fig 6. Kinesore treatment of mast cells affects microtubule structures independent of its effect on granule distribution. A)** RBL-2H3 cells were pretreated for 30 min with vehicle (DMSO), or 100 μM kinesore. Cells were left unstimulated, or antigen-stimulated for 30 min, fixed and stained for nuclei (blue), microtubules (green), or CD63+ granules (red). Granule distribution to the cell periphery is disrupted by kinesore treatment. Microtubule structures that normally project linearly to the cell periphery were disrupted by kinesore treatment. **B)** RBL-2H3 cells were pre-treated for 0 min, 15 min, 30 min or 60 min with kinesore then fixed (unstimulated), or antigen-stimulated for 30 min then fixed and stained for nuclei (blue) and microtubules (green). Microtubule linear structures were disrupted by kinesore after 30 min. Scale bar, 10 μm.

kinesore [30]. A time course of kinesore treatment showed that microtubule remodeling into looping structures in RBL-2H3 cells was not apparent after 15 min of incubation, while full disruption of the microtubule network into loops occurred by 60 min of incubation regardless of whether cells were stimulated or unstimulated (Fig 6B). Therefore, the loss of granule transport to the cell periphery precedes microtubule looping.

### Kinesore affects the association of granules with microtubule motors

Kinesin-1 is a microtubule motor protein that is activated by binding to cargo via cargo adaptor proteins that link vesicles, such as granules in activated mast cells, to microtubules for transport [46]. We examined the levels of expression of several lysosomal and secretory vesicle cargo adaptors in RBL-2H3 cells by quantitative PCR. This showed the expression of low levels of SKIP, Slp2 and Slp3 mRNA compared to GAPDH (Fig 7A). Previously, it was reported that Slp3 functioned as a secretory granule cargo adaptor in BMMCs [20], while SKIP functions as a cargo adaptor that is needed for the recruitment of kinesin-1 to lysosomes and melanosomes [28, 29, 47]. JIP3 and JIP4 were also detected, however they are associated with bidirectional vesicle transport in neurons and were not further examined [48]. Immunoblot of lysates from a variety of tissue culture cells showed abundant levels of the kinesin-1 heavy chain, Kif5b, and the presence of Slp3 immunoreactive bands, however SKIP could not be detected in RBL-2H3 cells (Fig 7B).

Kinesore activates the microtubule motor kinesin-1 in the absence of cargo binding by mimicking association with cargo adaptors [30]. Therefore, the loss of granule transport observed after kinesore treatment suggests that it may functionally inhibit the association of microtubule motors with cargo adaptors. To test this, we generated granule-enriched fractions from stimulated and unstimulated mast cells and probed them for the presence of Kif5b (kinesin-1 heavy chain) and the cargo adaptor Slp3. Granule-enriched fractions showed high levels of rat mast cell protease II (RMCP II) when prepared from unstimulated cells, and a 47% +/-7.2% reduction in RMCP II levels after stimulation for 30 min, likely due to exocytosis (Fig 7C, *DMSO*). Kif5b was also associated with granule-enriched fractions. Kif5b was recruited to granule-enriched fractions with levels increasing by 118% +/-17% when prepared from cells stimulated for 30 min (Fig 7C, *DMSO*). Kinesore treatment resulted in no recruitment of Kif5b to granule-enriched fractions but an increase in Slp3 levels (Fig 7C, *Kinesore*). There was no reduction in the levels of RMCP II in fractions prepared from kinesore-treated cells after stimulation, which indicates kinesore inhibited exocytosis. Therefore, kinesore likely inhibits exocytosis by blocking microtubule motor protein association with granules. Taken together, these results show that granules movement on microtubules tracks for exocytosis requires the activation of microtubule motors that bind to granule associated cargo adaptors.

## Discussion

Antigen-stimulation of mast cells induces cytoskeletal remodeling and granule transport to the plasma membrane resulting in exocytosis; also referred to as degranulation. Here we used live-cell imaging of RBL-2H3 cells to examine the dynamics of cytoskeletal remodeling and granule motility during antigen-stimulation. Actin remodeling at the surface of cells has been proposed to capture vesicles for exocytosis [49–51]. However, our results show actin mediated protrusions that form in response to mast cell stimulation are devoid of granules (see S1 Video via https://doi.org/10.6084/m9.figshare.19349552.v1). Compounds that depolymerize actin most often enhance exocytosis. This is congruent with a recent study using high resolution microscopy which showed cortical actin disassembly formed exocytosis zones at the surface of mast cells [52].

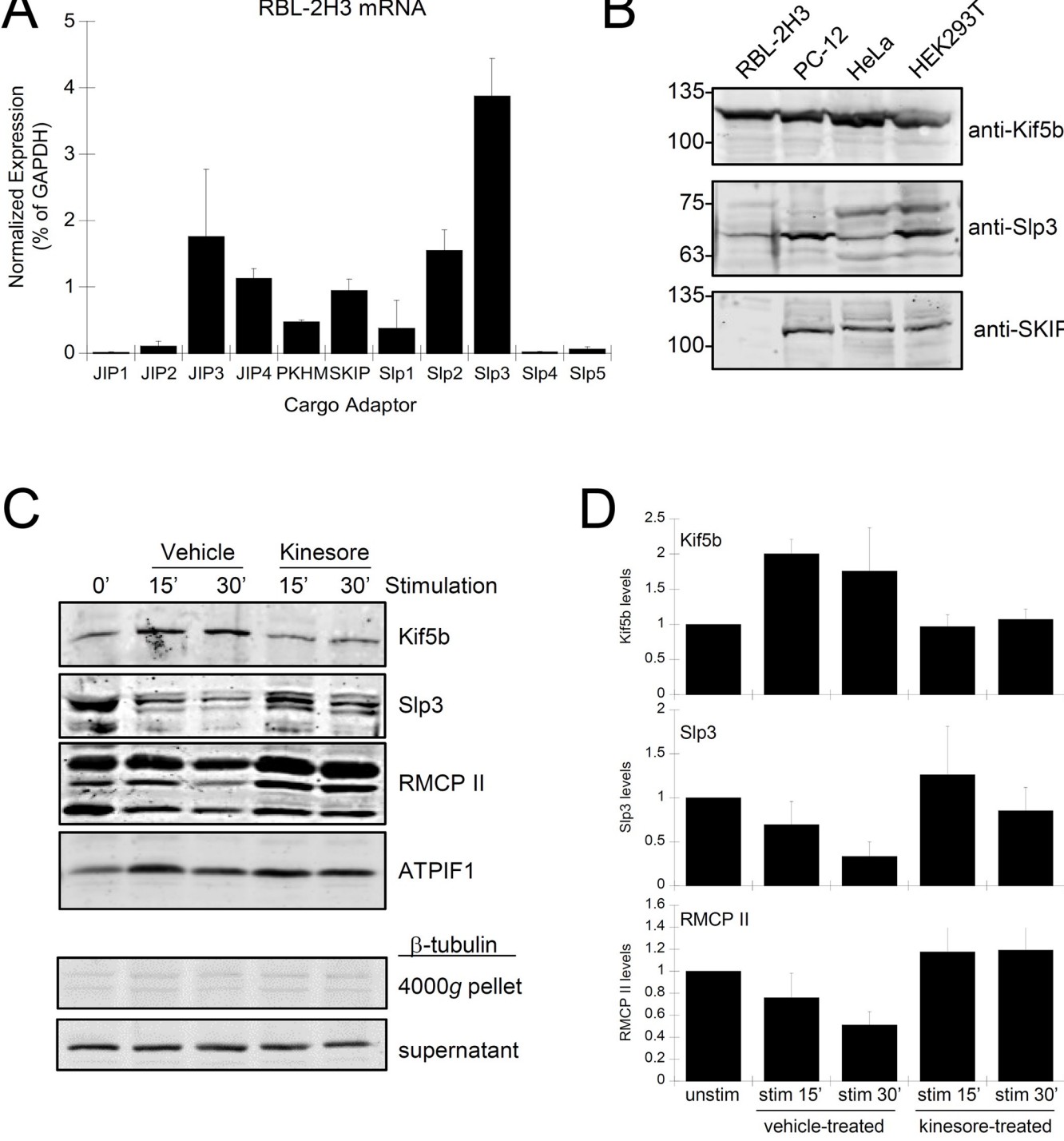

**Fig 7. Kinesore inhibits granule association of the microtubule motor kinesin-1. A)** Quantitative PCR analysis of mRNA isolated from RBL-2H3 cells. Gene expression level of cargo adaptors known to associate with secretory granules and lysosomes. **B)** Immunoblot of lysates from RBL-2H3, PC-12, HeLa and HEK293T cells ($10^7$ cells/ml lysate). Lysates were probed for Kif5b, and two cargo adaptors, Slp3 and SKIP. SKIP was not detected in RBL-2H3 cells. **C)** Granule-enriched fractions were prepared by differential centrifugation from unstimulated cells (*0'*), or from cells that were antigen-stimulated for 15 min and 30 min in the presence of kinesore or vehicle (0.5% DMSO). Fractions were probed by immunoblot for association of the microtubule motor subunit Kif5b, the cargo adaptor Slp3, the granule enzyme rat mast cell protease II (RMCP II) and the mitochondrial marker ATPIF1 (*upper panels*). Tubulin was not associated with the granule enriched fraction and instead was found in supernatant (*lower panels*). **D)** Levels of protein associated with granule-enriched fractions were examined by band densitometry of immunoblots. Values were normalized to unstimulated cells for each experiment (n = 3).

Our results support a role for microtubules in the transport and granule exocytosis mechanism. Our model for this part of the mechanism is that upon anti-stimulation, granules are recruited from the perinuclear region by association with microtubules to be driven to zones of exocytosis formed by cell spreading. This model is supported by the effect of microtubule depolymerizing agents which reduce mast cell degranulation and perturb granule motility and distribution (see S3 Video via https://doi.org/10.6084/m9.figshare.19349558.v1 and S4 Video via https://doi.org/10.6084/m9.figshare.19349564.v1) [17, 23]. However, the effect of these compounds could be due to gross perturbation of cell morphology since the formation of exocytosis zones has been shown necessary [7, 52]. Therefore, we tested the compound kinesore which has a more targeted effect on the function of the microtubule motor, kinesin-1 [30]. Kinesore-treated cells showed a lack of granule trafficking to the cell surface (see S6 Video via https://doi.org/10.6084/m9.figshare.19349573.v1) and also significant inhibition of granule exocytosis, both in primary mast cells (BMMCs) and in RBL-2H3 cells. Although kinesore also affected cell morphology with the formation of microtubule looping structures, which have been observed in other cell types treated with kinesore [30]. These structures formed progressively after 30 min of incubation while granules accumulated in the perinuclear region prior to the formation of microtubule loops. Therefore, kinesore-mediated inhibition of exocytosis seemed to be independent of the morphological effects on microtubule organization. To support this result we also examined the effects of reducing kinesin-1 function by RNAi-mediated knock-down of the kinesin-1 heavy chain subunit, Kif5b. Kinesin-1 knock-down showed near identical effects as kinesore; the reduction of exocytosis by ~50% and granule accumulation in the perinuclear region. The fact that both kinesore and kinesin-1 knock-down resulted in only a partial loss of exocytosis may be at least partially explain by granule heterogeneity with some granules poised for exocytosis and not needing an active transport event while others require recruitment from a reserve pool.

A growing body of research shows that microtubule nucleation is linked to mast cell activation [14, 27, 53, 54]. Such a mechanism would facilitate the regulation of granule exocytosis to cell activation. Our results indicate that the activation of granule transport for exocytosis is microtubule-based. This activation step is regulated by cargo adaptors that can dynamically associate with granules when antigen-stimulation signals are received [19]. The role of cargo adaptors is to link cargo (i.e. granules) to microtubule based motors. Therefore, in mast cells, granule association with cargo-binding adaptors facilitates their association with microtubule based motors to drive the degranulation process [20]. We found kinesin-1 was recruited to a granule enriched fraction during antigen stimulation and its levels on granules were reduced by kinesore. Kinesore is known to activate kinesin-1 motor function in the absence of cargo by binding to the cargo adaptor site. Therefore, it is likely that kinesore inhibits the coupling of a granule cargo adaptor and kinesin-1. We also observed a lack of persistent granule motility in the direction of microtubule growth. There remained a large pool of static granules which suggest that the cargo adaptor may be a limiting factor in the activation for motility. Periodically, we also observed granules that moved in the opposite direction. Granules have been shown to associate with dynein motors resulting in transport towards microtubule minus ends [55]. This mechanism negatively regulates exocytosis and may facilitate the ability to abruptly shut down exocytosis and deactivate of mast cells, affording more control on inflammation.

Cargo-binding adaptor recruitment is likely a pivotal regulatory step for the activation of granule transport and exocytosis. We detected the expression of numerous cargo adaptors in RBL-2H3 cells and previously it was shown that Slp3 functions as a granule cargo adaptor in BMMCs and can link to kinesin-1 through a PI3 kinase-activated mechanism [20]. SKIP (PLEKHM2) was the cargo adaptor targeted by kinesore [30]. SKIP was previously implicated

in kinesin-1 activation during salmonella infection [56]. Recent studies have defined the role of SKIP in lysosomal transport and kinesin-1 activation. SKIP has been shown to serve as a linker protein between lysosomal membrane proteins and the kinesin-1 motor [28, 29]. SKIP plays a vital role in microtubule plus end directed motility of lysosomes and the lysosome related organelle, melanosomes [28, 47]. Hence, we consider SKIP may be a potential cargo-binding adaptor for mast cell secretory granules.

The molecular machinery involved in motor protein-cargo binding is highly heterogeneous. In melanocytes, SKIP is involved in the transport of both lysosomes and melanosomes [47]. It differentiates between these organelles via binding to different membrane associated-cargo proteins, binding to Arl8 on lysosomes whiles binding to Rab1a on melanosomes. In hematopoietic cells, cargo adaptors can mediate the transport of a variety of granules via binding to different membrane associated-cargo proteins. In cytotoxic T lymphocytes Slp3 facilitates kinesin-1 based transport of terminal lytic granules via binding to Rab27a on lytic granules and to the kinesin-1 motor protein [19]. Yet in murine BMMCs, it has been demonstrated that Slp3 enables kinesin-1 dependent transport of CD63+ granules via binding to Rab27b [20]. Therefore, an important future study is to determine the mechanism that activates mast cell cargo-binding adaptors in their recruitment to granules and coupling with the kinesin-1 microtubule motor.

## Supporting information

**S1 Fig. Analysis of EB3-puncta after kinesore treatment.** Confocal images from live-cell imaging of microtubule dynamics and granule movement of mast cells treated with 100 μM kinesore (*see S6 Video via* https://doi.org/10.6084/m9.figshare.19349573.v1). RBL-2H3 cells were transfected with EB3-tdTomato to label nascent microtubules and incubated with Lyso-tracker green to label granules. Cells were imaged for 1 min, then antigen-stimulated and concurrently kinesore was added, followed by 15 min of imaging. Images were extracted from the green channel at 5, 10 and 15 min time points. The intensity threshold was set to 1% and particles between 4–50 square pixels were counted in ImageJ.
(PDF)

**S2 Fig. LysoTracker labels CD63-positive vesicles in RBL-2H3 mast cells.** Live RBL-2H3 cells were labelled with LysoTracker Red (ThermoFisher). Cells were then left unstimulated (*A*), or antigen-stimulated for 20 min (*B*). Cells were fixed and CD63-positives vesicles were immuno-labelled with monoclonal CD63 antibodies (clone AD1, BioRad). F-actin was labelled with phalloidin iFluor-405 (Abcam). Images were taken with a Zeiss Observer Z1 epifluorescence microscope using a 63X 1.4 NA objective. Bottom panels show zoomed images of area indicated in the upper panels. Note that CD63 antibodies label many outlier vesicles that are not labelled with LysoTracker Red (*arrows*), while LysoTracker Red labelled vesicles predominately overlap with CD63 labelling. Scale bar, 10 μm.
(PDF)

**S3 Fig. Analysis of colocalization of LysoTracker and CD63 labels.** Colocalization of Lyso-Tracker and CD63 labels in RBL-2H3 cells, as indicated in Figure Legend S3. Cells were imaged with a Leica UltraVIEW VoX spinning-disk confocal microscopy using a 63X, 1.4 NA objective. Colocalization of LysoTracker Red and CD63 signals was determined by calculation of Pearson's correlation coefficient using the Volocity 6.1 plugin software. Seven images with at least 2 cells per image were analyzed; shown are representative analyses of unstimulated cells (*A*) and stimulated cells (*B*). Average Pearson's correlation coefficients (*r*) were 0.714 +/-0.033 for unstimulated cells and 0.622 +/-0.055 for stimulated cells. Coefficients above 0.7 are

generally considered strong correlation. Correlation is reduced during stimulation due to exocytosis which reduces LysoTracker labelling but not CD63.
(PDF)

**S1 Raw images. Unaltered images for blots presented in Figs 5 and 7.** See the legend for Figs 5A and 7B and 7C for details.
(PDF)

**S1 File. Raw data for results presented in figures and Table 2.** See the figure legend for Figs 2A, 2E, 3A, 5B and 7A and 7D for details.
(PDF)

## Acknowledgments

We thank Dr. Andrew Simmonds for technical support with live-cell imaging and Dr. Dean Befus for rat mast cell protease II antibodies.

## Author Contributions

**Conceptualization:** Jeremies Ibanga, Gary Eitzen, Yitian Guo.

**Formal analysis:** Eric L. Zhang.

**Funding acquisition:** Gary Eitzen.

**Investigation:** Jeremies Ibanga, Eric L. Zhang, Gary Eitzen, Yitian Guo.

**Methodology:** Jeremies Ibanga, Eric L. Zhang, Gary Eitzen, Yitian Guo.

**Supervision:** Gary Eitzen, Yitian Guo.

**Validation:** Yitian Guo.

**Writing – original draft:** Jeremies Ibanga, Gary Eitzen, Yitian Guo.

**Writing – review & editing:** Jeremies Ibanga, Gary Eitzen, Yitian Guo.

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
