## [Decision Letter · Decision Letter 0]

9 Sep 2021

PONE-D-21-21940Mast cell granule motility and exocytosis is driven by dynamic microtubule formation and kinesin-1 motor functionPLOS ONE

Dear Dr. Eitzen,

Thank you for submitting your manuscript to PLOS ONE. After careful consideration, we feel that it has merit but does not fully meet PLOS ONE’s publication criteria as it currently stands. Therefore, we invite you to submit a revised version of the manuscript that addresses the points raised during the review process.

1. Both Reviewers expressed their concerns regarding specificity of the kinesore. The dose-dependent effect of this pharmacological agent should be established. Alternatively,  the authors should demonstrate the role of kinesin-1 using additional approaches such as kinesin-1 knockdown.

2. The authors should demonstrate that the Lyso tracker colocalizes with CD63, a marker for the secretory granules.

3. The authors should develop and use the method for image analyses. It is absolutely required that all data including images are analyzed using statistical approaches in order to meet the PlosOne technical standards.

4. The authors should revise the method and discussion as recommended by the Reviewer 1. The authors should draw conclusions from data presented without extensive overinterpreting the data.

We look forward to receiving your revised manuscript.

Kind regards,

Yulia Komarova

Academic Editor

PLOS ONE

Journal Requirements:

Jeremies Ibanga is the recipient of an Undergraduate Summer Research Award (USRA) from The Natural Sciences and Engineering Research Council of Canada (NSERC). Yitian Guo is the recipient a studentship from the China Scholarship Council. This work was supported by a Discovery Grant from The Natural Sciences and Engineering Research Council of Canada (NSERC) grant number RGPIN-2019-05466 to Gary Eitzen. 

J. I. is the recipient of an Undergraduate Summer Research Award (USRA) from The Natural Sciences and Engineering Research Council of Canada (https://www.nserc-crsng.gc.ca/index_eng.asp). Y. G. is the recipient a studentship from the China Scholarship Council (https://www.chinesescholarshipcouncil.com). This work was supported by a Discovery Grant from The Natural Sciences and Engineering Research Council of Canada grant number RGPIN-2019-05466 to Gary Eitzen. The funders had no role in study design, data collection and analysis, decision to publish, or preparation of the manuscript.

Reviewers' comments:

Reviewer's Responses to Questions

**Comments to the Author**

1. Is the manuscript technically sound, and do the data support the conclusions?

Reviewer #1: Partly

Reviewer #2: Partly

2. Has the statistical analysis been performed appropriately and rigorously? 

Reviewer #1: Yes

Reviewer #2: Yes

3. Have the authors made all data underlying the findings in their manuscript fully available?

Reviewer #1: Yes

Reviewer #2: Yes

4. Is the manuscript presented in an intelligible fashion and written in standard English?

Reviewer #1: Yes

Reviewer #2: Yes

5. Review Comments to the Author

Reviewer #1: The authors use small molecule inhibitors of actin, microtubules, and a new kinesin inhibitor to study the role of microtubules and kinesin-1 in in mast cell motility and exocytosis. Fixed and live single cell imaging, cell fractionation, an exocytosis assay and PCR were used. Authors conclude that exocytosis of mast cell granules involves kinesin-1 on microtubules.

Major points-

Fig 1 A and other Figs- Most granules do not move persistently in one direction in manner that is consistent with microtubules. Do the mean granule velocities include the granules that do not move? If so, the mean velocity would be much lower. If non-moving granules are not included in the mean velocity calculations, then what is the exclusion criteria for granules? Selection/Exclusion criteria should be included in Methods section.

As a broader point, the lack of persistent directional movement needs incorporated into the discussion and model of the proposed transport mechanism.

Fig 2 – A, E- shows that Nocodazole decreases % exocytosis, but not granule speed, and paclitaxel has little effect on % exocytosis but decreased granule speed. Possible reasons for this observation should be included in the Discussion section.

Fig 2 B, C- many granules appear outside of cell boundary, while in Fig2 D only inside of cell boundary. The authors should elaborate on the reason or cause.

Fib 3 B, C, D- The author interpretations from the images is that kinesore effects granule distribution but not EB3 or F-actin. The images do not clearly support these conclusions without a quantitative image analysis approach. An image analysis approach is needed.

Kinesore is a new inhibitor, not yet widely tested. The specificity of the inhibitory compound is questionable at concentrations used (100 uM). At these high concentrations, off-target effects are likely to occur. A more precise, and established approach is kinesin-1 knockdown. Additionally, live cell imaging of microtubules and granules would be valuable.

Minor points-

Line 84 “Mast cell granule...” needs corrected.

Line 254- “Hence granules moved at velocities similar to the rate of microtubule growth suggesting that granules are driven on new microtubules” Similar velocities is not strong supporting evidence for the statement. Suggest revision.

Lines 773 and 781- Movie 2 and 5 links not working

Reviewer #2: The report by Ibanga et al. investigates the role of the microtubule motor protein kinesin-1 and microtubule remodeling in mast cell (MC) degranulation. The authors show using live-cell imaging that de novo microtubule formation coordinated granule transport. Functionally, they used the kinesore drug, described to activate kinesin-1 in absence of cargo, to show a defective granule translocation on microtubule and secretion. They conclude that granules are driven by kinesin-1 on microtubule to facilitate degranulation.

The authors revisit published studies on already described molecular mechanisms that regulate MC degranulation.

The manuscript is well written and easy to follow. However, I have some concerns that need to be addressed.

1) We can regret that the majority of the study is only done on RBL-2H3 cell line and not on primary MC (BMMC).

2) The authors used Lyso Tracker to follow secretory granules. The authors should show that in this condition the Lyso tracker colocalizes with CD63, a secretory granule marker.

3) It seems that kinesore may impact microtubule dynamic independently of its effect on kinesin-1. The authors observed that the effect of kinesore on microtubule remodeling into looping structure occurs after the loss of granule transport. However, it is not so clear whether kinesore may impact microtubule dynamic by influencing post-translational modifications of the microtubule tracks (acetylation, detyrosination, etc..) that can indirectly impact kinesin-1 functionality. In addition, do the authors know whether kinesore may impact functionality of other members of kinesin family? The lack of specificity of kinesore could be the major problem to precisely understand which molecular mechanism is impacted to explain the MC degranulation defect observed.

6. PLOS authors have the option to publish the peer review history of their article (what does this mean?). If published, this will include your full peer review and any attached files.

Reviewer #1: No

Reviewer #2: No

---

## [Author Response · Author response to Decision Letter 0]

1 Dec 2021

Response to editor’s comments

1. Both Reviewers expressed their concerns regarding specificity of the kinesore. The dose-dependent effect of this pharmacological agent should be established. Alternatively, the authors should demonstrate the role of kinesin-1 using additional approaches such as kinesin-1 knockdown.

RESPONSE: We provide a dose-response curve for kinesore in Figure 3A. However, we do agree that additional validation of this result would significantly strengthen the manuscript. Hence we also generated mast cell Kif5b knock down strains (Kif5b is the heavy chain of kinesin-1). In the new Figure 5 we characterized the effect. Exocytosis was significantly reduced and granules were retained in a perinuclear region; this result is very similar to the effect of kinesore and hence supportive of the conclusions drawn. 

2. The authors should demonstrate that the Lyso tracker colocalizes with CD63, a marker for the secretory granules.

RESPONSE: We performed a colocalization study of LysoTracker and anti-CD63 labeling in resting and stimulated RBL-2H3 cells. This is included as Figures S2 and Figure S3, supplemental data. Peason’s correlation coefficients were calculated for these two cellular states which showed >0.7 in the resting state and slightly lower in the stimulated state. A drop in the correlation coefficient is expected after stimulation since exocytosis would result in the loss of LysoTracker signal (requires an acidified organelle which would be neutralized upon exposure to HTB during exocytosis. It is important to note that many studies have used LysoTracker to label mast cell granules while other have used CD63 (we referenced a few). To our knowledge, this study is the first that directly examines their colocalization.

3. The authors should develop and use the method for image analyses. It is absolutely required that all data including images are analyzed using statistical approaches in order to meet the PlosOne technical standards.

RESPONSE: We have included a section in the Materials and Methods that describes the analysis of images and in the reported results we include statistical analyses. Granule motility was analyzed in live-cell images (Figures 1-3). In addition, in Figure 3 we concluded that the drug kinesore does not affect de novo formation of microtubules based on EB3-GFP puncta. S1 Fig in supporting information shows this result. 

Imaging in Figure 4 and Figure 5C we show effects on granule distribution. Images show granules are perinuclear when cells are unstimulated. Stimulation normally results in granule redistribution throughout the cell. However, microtubule drugs or Kif5b knock-down blocks this redistribution. Profile plots are included in these figures to reinforce this result. Additionally, we quantified this result using defined regions of interest using an established protocol outlined in the methods and the results detailed and statistically analyzed as presented in Table 2. Note that we quantified this result for stimulated cells only since unstimulated cells all show very similar perinuclear granule distribution.

Some imaging results are simply not amenable to statistical analysis since they do not generate numerical results. In particular, the microtubule “looping” induced by kinesore. This effect was first characterized by Randall et al., 2017 (PNAS) and again used by Paul et al., 2021 (JCB). No methods to generate a numerical analysis of this looping effect were provided. Our results in Figure 6 show the effect of kinesore on microtubule morphology is unambiguously 100% penetrant. 

4. The authors should revise the method and discussion as recommended by the Reviewer 1. The authors should draw conclusions from data presented without extensive over-interpreting the data.

RESPONSE: Many additions have been made to the methods. In particular, we added an Image Analysis section to detail these additional methods used in the results. We have toned down our interpretation of the data and removed statements where we were potentially extending our interpretation of the data (this can be seen in the redlined document).

 

Point-by-point response to reviewers’ comments

Reviewer #1 major points:

Fig 1 A and other Figs- Most granules do not move persistently in one direction in manner that is consistent with microtubules. Do the mean granule velocities include the granules that do not move? If so, the mean velocity would be much lower. If non-moving granules are not included in the mean velocity calculations, then what is the exclusion criteria for granules? Selection/Exclusion criteria should be included in Methods section.

RESPONSE: We have provided specific details in the methods on how granules are included, or excluded, in the velocity calculations. Some granules do not move; these are mainly concentrated in the perinuclear region and our interpretation is that they have not been activated. Our data support and activation mechanism whereby a cargo adaptor binds the granule and recruits the motor. This does not occur for 100% of the granules.

As a broader point, the lack of persistent directional movement needs incorporated into the discussion and model of the proposed transport mechanism.

RESPONSE: We have added to the discussion the points raise here. Our proposed model of the transport mechanism is incorporated into the discussion. How our data support this model is discussed in detail. We also explain within this model our interpretation of why persistent movement is not observed for all granules, such as backwards movement of granules (new reference 55).

Fig 2 – A, E- shows that Nocodazole decreases % exocytosis, but not granule speed, and paclitaxel has little effect on % exocytosis but decreased granule speed. Possible reasons for this observation should be included in the Discussion section.

RESPONSE: We have included discussion of the paradoxical effects of nocodazole and paclitaxel in the results section rather than the discussion which is highly focused on our proposed model of microtubule regulated exocytosis.

Fig 2 B, C- many granules appear outside of cell boundary, while in Fig2 D only inside of cell boundary. The authors should elaborate on the reason or cause.

RESPONSE: This is an important oversight that we have clarified. The granules that appear to be outside of a cell boundary are in fact inside a cell that has not been transfected with the GFP probes.

Fib 3 B, C, D- The author interpretations from the images is that kinesore effects granule distribution but not EB3 or F-actin. The images do not clearly support these conclusions without a quantitative image analysis approach. An image analysis approach is needed.

RESPONSE: We have included a quantification of granule distribution and EB3-GFP puncta in kinesore-treated cells and quantification of granule distribution in microtubule drug-treated cells. However, EB3-GPF puncta were not quantified in microtubule drug-treated cells since there were none.

The dynamics of F-actin formation after stimulation is unfortunately not amenable to direct quantitative analysis. Our data in Fig 4 we feel quite clearly shows well spread cells and lamellipodia formation in vehicle-treated cells, and comparably so in kinesore-treated cells. In addition, F-actin dynamics as observed in S1 Video is directly comparable to that in kinesore-treated cells shown in S7 Video. Hence our conclusion that kinesore does not affect F-actin formation we feel is justified.

Kinesore is a new inhibitor, not yet widely tested. The specificity of the inhibitory compound is questionable at concentrations used (100 uM). At these high concentrations, off-target effects are likely to occur. A more precise, and established approach is kinesin-1 knockdown. Additionally, live cell imaging of microtubules and granules would be valuable.

RESPONSE: We have established a kinesin-1 knock-down strain using Kif5b shRNAs (Kif5b is the kinesin-1 heavy chain). The analysis of this strain is provided in a new figure (Fig 5). The results support the conclusions from the effects of kinesore treatment.

Reviewer #1 minor points:

Line 84 “Mast cell granule...” needs corrected.

RESPONSE: The grammar here has been corrected

Line 254- “Hence granules moved at velocities similar to the rate of microtubule growth suggesting that granules are driven on new microtubules” Similar velocities is not strong supporting evidence for the statement. Suggest revision.

RESPONSE: Our potential over-interpretation of the data here has been modified to “Hence granules moved at velocities similar to the rate of microtubule growth which may facilitate coordination between granules motility and de novo microtubule formation”. The idea is that if granules moved faster than microtubule formation it would not allow coordination since microtubule tracks would not to be present.

Lines 773 and 781- Movie 2 and 5 links not working

RESPONSE: We have tested the Movie links (renamed S# Videos) and they work in our hands. The URL may need to be copy/pasted to work in different browsers. We have indicated this.

Reviewer #2 major concerns:

1) We can regret that the majority of the study is only done on RBL-2H3 cell line and not on primary MC (BMMC).

RESPONSE: We tested the new drug kinesore on BMMCs. However, BMMCs are not highly amenable to live cell imaging which was our major approach in this study. BMMCs are non-adherent and very small making it difficult to analyze effects on granule motility and distribution.

2) The authors used Lyso Tracker to follow secretory granules. The authors should show that in this condition the Lyso tracker colocalizes with CD63, a secretory granule marker.

RESPONES: We performed a LysoTracker-anti-CD63 colocalization study (S3 Fig). Calculation of Pearson’s correlation coefficient suggest strong colocalization prior to stimulation which decreases slightly after stimulation (S4 Fig). We explain that this decrease would be expected due to secretion which would affect acidic levels of vesicles and LysoTracker staining but not anti-CD63 labelling.

3) It seems that kinesore may impact microtubule dynamic independently of its effect on kinesin-1. The authors observed that the effect of kinesore on microtubule remodeling into looping structure occurs after the loss of granule transport. However, it is not so clear whether kinesore may impact microtubule dynamic by influencing post-translational modifications of the microtubule tracks (acetylation, detyrosination, etc..) that can indirectly impact kinesin-1 functionality. In addition, do the authors know whether kinesore may impact functionality of other members of kinesin family? The lack of specificity of kinesore could be the major problem to precisely understand which molecular mechanism is impacted to explain the MC degranulation defect observed.

RESPONSE: Kinesore binds kinesin-1 and so it is unlikely to affect microtubule modifications. It would be quite interesting to identify the mechanism of microtubule looping associated with kinesore treatment but this is beyond the scope of our study. At this point it is unknown whether kinesore impacts other microtubule motors; we feel this is also beyond the scope of this work. However, to address the concern of kinesore specificity we generated a kinesin-1 knock-down strain using Kif5b shRNAs. The analysis of this strain is included in a new figure (Fig 5). The results confirm that kinesin-1 is required for granule redistribution from the perinuclear region and for exocytosis.

---

## [Decision Letter · Decision Letter 1]

24 Feb 2022

Mast cell granule motility and exocytosis is driven by dynamic microtubule formation and kinesin-1 motor function

PONE-D-21-21940R1

Dear Dr. Eitzen,

We’re pleased to inform you that your manuscript has been judged scientifically suitable for publication and will be formally accepted for publication once it meets all outstanding technical requirements.

Kind regards,

Yulia Komarova

Academic Editor

PLOS ONE

Additional Editor Comments (optional):

Reviewers' comments:

Reviewer's Responses to Questions

**Comments to the Author**

1. If the authors have adequately addressed your comments raised in a previous round of review and you feel that this manuscript is now acceptable for publication, you may indicate that here to bypass the “Comments to the Author” section, enter your conflict of interest statement in the “Confidential to Editor” section, and submit your "Accept" recommendation.

Reviewer #1: All comments have been addressed

Reviewer #2: All comments have been addressed

2. Is the manuscript technically sound, and do the data support the conclusions?

Reviewer #1: Yes

Reviewer #2: Yes

3. Has the statistical analysis been performed appropriately and rigorously? 

Reviewer #1: Yes

Reviewer #2: Yes

4. Have the authors made all data underlying the findings in their manuscript fully available?

Reviewer #1: Yes

Reviewer #2: Yes

5. Is the manuscript presented in an intelligible fashion and written in standard English?

Reviewer #1: Yes

Reviewer #2: Yes

6. Review Comments to the Author

Reviewer #1: All the comments have been adequately addressed and the manuscript is now acceptable for publication.

Reviewer #2: The authors have answered most of my questions thus I recommend the paper for publication.

7. PLOS authors have the option to publish the peer review history of their article (what does this mean?). If published, this will include your full peer review and any attached files.

Reviewer #1: **Yes: **Joseph M Schober

Reviewer #2: No

---

## [Editor Report · Acceptance letter]

8 Mar 2022

PONE-D-21-21940R1 

Mast cell granule motility and exocytosis is driven by dynamic microtubule formation and kinesin-1 motor function 

Dear Dr. Eitzen:

I'm pleased to inform you that your manuscript has been deemed suitable for publication in PLOS ONE. Congratulations! Your manuscript is now with our production department. 

Kind regards, 

on behalf of

Dr. Yulia Komarova 

Academic Editor

PLOS ONE